REGISTERED REPORT PROTOCOL

# Validation and applicability of the music ear test on a large Chinese sample

**Xiaoyu Wang[1], Xiubo Ren[1], Shidan Wang[1], Dan Yang[1], Shilin Liu[1], Meihui Li[1], Mingyi Yang[1], Yintong Liu[1], Qiujian Xu** [1,2]*

1 Music College, Catholic University of Daegu, Gyeongsan-si, Gyeongsangbuk-do, Rep. of Korea, 2 School of Arts and Design, Yanshan University, Qinhuangdao, China

* xuqiujian@ysu.edu.cn

**Data Availability Statement:** All relevant data are within the manuscript and its Supporting

## Abstract

In the context of extensive disciplinary integration, researchers worldwide have increasingly focused on musical ability. However, despite the wide range of available music ability tests, there remains a dearth of validated tests applicable to China. The Music Ear Test (MET) is a validated scale that has been reported to be potentially suitable for cross-cultural distribution in a Chinese sample. However, no formal translation and cross-cultural reliability/validity tests have been conducted for the Chinese population in any of the studies using the Music Ear Test. This study aims to assess the factor structure, convergence, predictiveness, and validity of the Chinese version of the MET, based on a large sample of Chinese participants (n≥1235). Furthermore, we seek to determine whether variables such as music training level, response pattern, and demographic data such as gender and age have intervening effects on the results. In doing so, we aim to provide clear indications of musical aptitude and expertise by validating an existing instrument, the Music Ear Test, and provide a valid method for further understanding the musical abilities of the Chinese sample.

## Introduction

### Importance of musical ability

In the realm of interdisciplinary inquiry, researchers worldwide have directed their attention toward the significance of musical ability. Notably, the impact of musical proficiency on the genesis and progression of diverse capabilities, including intelligence [1], language aptitude [2, 3], digit span [4], and working memory [5], has undergone extensive examination. This body of research substantially contributes to a profound comprehension of the pivotal role played by musical ability in the course of human development. The imperative for objectively valid and standardized tests of musical ability becomes evident, serving as a crucial foundation for conducting such investigations and establishing quantitative benchmarks for categorizing musical proficiency within sampled populations.

In pursuit of an appropriate music ability test for application in the Chinese population, the researchers conducted a comprehensive review of extant tests (refer to Table 1). During this review, potential limitations of current assessments were identified. Some tests were designed

Information files. All the data and resources covered in the article can be found on the Centre for Open Science (https://osf.io/kxey3/?view_only= e97871f8520347279c807022e29e381d).

**Funding:** The authors received no specific funding for this work.

**Competing interests:** The authors have declared that no competing interests exist.

with a narrow focus on specific sample groups [6, 7], while others were released without undergoing thorough reliability and validity testing or concentrated solely on excluding specific deficits in musical ability [7, 8].

Furthermore, certain assessments lacked direct evidence affirming their cross-cultural validity [9, 10]. Some failed to align with the requisites of online testing platforms, thereby impeding mass distribution [6]. Additionally, a subset of tests remained inaccessible to the public or was characterized as outdated. This critical evaluation of existing instruments underscores the need for a culturally appropriate and methodologically sound music ability test for the Chinese population.

Among the various scales available, the Music Ear Test (MET) is considered the most suitable research instrument for this study. Several factors support this choice. Firstly, the MET has garnered widespread adoption and undergone repeated validation in subsequent studies. It serves as a primary indicator of musical ability and has been systematically compared and correlated with diverse constructs, including personality type [11], openness to experience [12], intelligence [1], language ability [13], cross-cultural music listening adaptation [14], musical reward effect [15], and episodic transfer of musical ability [16].

Secondly, the MET has demonstrated its utility across various countries, suggesting its potential suitability for cross-cultural distribution within a Chinese sample [13, 14]. Thirdly, the MET exhibits a well-organized structure (refer to Table 1), and subsequent studies consistently report high reliability data (refer to Table 2). Lastly, the validity of the online version of the MET has been substantiated [17], further affirming its appropriateness for this research endeavor.

**Table 1. Overview of the music ability test.**

| Assessment scale, author | Age(average) | Content | Format | Reliability | Validity | Cross-cultural validity |
|---|---|---|---|---|---|---|
| Montreal Battery of Evaluation Musical Abilities (MBEMA) [7] | Ages 6–8 | Contour, interval, scale, rhythm, and memory tests. | Computer loudspeaker and verbal answers | Not reported | Not reported | Yes, Beijing sample (N = 91) |
| Music Cognitive Test (MCT) [6] | Ages≥18 | Test items and observation items | Paper and pencil test | Internal (Cronbach's alpha):.84-.91, factor analyses, sensitivity and specificity, inter-rater reliability, adjustment scoring coefficient | Content validity, concurrent validity, test-retest | Yes, Italian and English versions |
| Profile of Music Perception Skills (PROMS) [10] | Adult population | Tonal, qualitative, temporal, dynamic | Headphone (MP3) and paper answer sheet | Internal consistency and test-retest (r. = .85) | Convergent validity | No |
| Asztalos & Csapó (2017) [9] | Student from the 1st to 11th grades | Basic musical hearing abilities (rhythm, tempo, melody, pitch, harmony, timbre, dynamics) | Computer labs and online answer sheet | Internal (Cronbach's alpha):.86 | Not reported | No |
| Adaptive Music Perception Test [8] | Hearing impaired | Music perception with hearing loss (meter, harmony, melody, timbre) | Computer-driven test | Discrimination thresholds | Test-retest | No |
| Music Ear Test (MET) [23] | Adult population | Melody<br>rhythm | WAV and MP3 | Internal (Cronbach's alpha):.69-.85 | Convergent validity and criterion validity | Have used in tonal language sample (Swaminathan et al., 2021) |

*Note.* The information reported in the table reflects that provided in the original research.

**Table 2. The demographics of the pre-experimental participants and the grouping of the participants.**

| ITEMS | | GROUP | | | | Total |
|---|---|---|---|---|---|---|
| | | **Musician** | **Amateur** | **Non-musician** | **Unskilled musician** | |
| AGE (years old) | Under 18 | - | - | 2 | 1 | 3 |
| | 18~24 | 3 | 7 | 58 | 8 | 76 |
| | 25~30 | 1 | - | - | 3 | 4 |
| Gender | MALE | 2 | 3 | 38 | 3 | 46 |
| | FEMALE | 2 | 4 | 22 | 9 | 37 |
| Professional music education experiences | | YES | NO | NO | YES | - |
| Number of music skills | | 3–6 | 1 | 0 | 1–2 | - |
| Academic Qualifications | Senior secondary | - | - | 1 | - | 1 |
| | Specialist degree | | | 1 | | 1 |
| | Bachelor's degree | 3 | 7 | 58 | 12 | 80 |
| | Master's degree and above | 1 | - | - | - | 1 |
| ResponseMethod | Online | 3 | 1 | 18 | 6 | 28 |
| | offline | 1 | 6 | 42 | 6 | 55 |
| SUM | | 4 | 7 | 60 | 12 | 83 |

## Definition of musical ability

Musical ability is a multifaceted construct [18] encompassing both an inherent auditory neurobiological foundation, representing innate music perception ability, and a cultivated, natural/spontaneous development involving exposure to musical culture and musical training [19]. In the early stages of research, scholars delineated musical ability into two distinct realms, each operating independently. Specifically, musical aptitude denotes the "potential for learning music" [20] before formal training and accomplishment. In contrast, music expertise, also known as music skills, musical training, musical competence, or music achievement, manifests as the outcome of purposeful practice [21] or the subtle influence of musical culture. Consequently, individuals with high musical aptitude and those who have mastered specific musical skills are often regarded as a homogeneous group [11, 22, 23].

Consequently, prior examinations of musical aptitude tests have categorized them into two principal sections: assessments of inherent musical aptitude and gauges of proficiency in musical skills. Music achievement tests specifically evaluate the skill levels acquired by groups of professionally trained musicians. A binary taxonomy, predicated on self-reported learning experiences, duration, and skill acquisition in music, has emerged as a prevalent early music achievement test in research. Recognized for its simplicity and efficacy, this method provides a valid benchmark for appraising music skill acquisition [5]. Subsequent developments in music achievement tests have built upon this basic dichotomy, refining classifications and prerequisites for self-reported music skills, thereby facilitating more accurate assessments of general music achievement [22, 24–26]. These tests essentially scrutinize past practice experiences.

In contrast, musical ability tests diverge from the context of musical training experiences and cast a broader focus on assessing the musical potential inherent in individuals.

## Beyond the binary: A nuanced understanding of musical ability

Recent findings challenge the traditional binary definition of musical ability, highlighting its intricate nature shaped by a multitude of interactions between genetic predispositions and environmental factors [16, 27]. Musical ability, as indicated by Swaminathan and Schellenberg (2018) [18], is inherently compound. On one hand, Mankel and Bidelman (2018) [19]

conducted a physiological cognitive experimental paradigm, revealing that formal musical experiences are neither essential nor sufficient conditions for enhancing neural coding and sound perception in the brain. Even informal or brief musical exposure can contribute to the development of individual musical aesthetics, melodic rhythm recognition, and other musical abilities [28]. On the other hand, research suggests that while deliberate practice is a crucial factor for achieving musical proficiency [21], this link is neither unique nor decisive [29]. Individuals with inherent musical aptitude or talent may yield more significant practice results [30]. These nuanced findings transcend the conventional categorization of musicians and non-musicians, introducing the concepts of the 'unskilled musician' (musically untalented music learner) and the 'music sleeper' (extraordinarily musically talented non-professional learner).

Simultaneously, it underscores inadequacies in existing scales in capturing the full extent of the distinction between innate and experience-dependent effects [19]. Existing scales may overlook pre-existing differences in musical ability within individuals lacking formal musical training, leading to confusion between gifted musicians without training and music learners without inherent musical ability. This deficiency prevents precise quantification of specific musical abilities (e.g., tempo, pitch, rhythm, timbre, melody perception, or their combinations), thereby posing limitations in research [10, 11, 16].

Therefore, the expansion of subjects related to musical ability, elucidation of the specific orientation of the results in a musical ability test, and the degree of differentiation are pivotal indicators for the success of a comprehensive musical ability test.

Originally designed to explore musical competence, the Music Ear Test (MET) has demonstrated robust group separation based on the frequency of current weekly musical practice/playing [4, 23]. Numerous subsequent studies have embraced this criterion, considering it a key variable linked to contemporary music acquisition [1, 4, 14, 31]. However, as the concept of musical ability evolved, questions arose regarding the MET's initial purpose. Correia et al. (2022) [17] scrutinized MET results among non-musicians, discovering that a noteworthy proportion of untrained individuals (12%) outperformed professional musician participants. Additionally, exceptionally talented non-musicians (top 25% decile) exhibited heightened rhythmic proficiency. Some researchers have directly referred to MET as a term denoting music aptitude, while others have overlooked its broader musical ability-related attributes, treating it merely as a test for musical pitch and rhythmic discrimination, emphasizing auditory discrimination ability [1, 11, 16, 32].

Given the inconclusive directionality of MET results, the present study will consider informal musical practices and music-related inclinations [11]. The study aims to elucidate the directionality of MET results through a refined categorization of the musical subject group. Moreover, the incomplete and non-standardized validation methods have impeded the dissemination and advancement of MET. Despite numerous validations of its reliability and validity (see Table 3), data pertaining to its Factorial Structure remain unreported.

**Table 3. Anticipated internal consistency reliability statistics (Cronbach's alpha) for scores on the MET and anticipated sample size.** For comparison purposes, values from four previous reports are provided.

| Item | Melody | Rhythm | Total | Sample Numbers |
|---|---|---|---|---|
| Wallentin et al., 2010 [23] | .82 | .69 | .85 | N = 60 |
| Swaminathan et al., 2021 [16] | **.73** | **.62** | **.78** | N = 523 |
| Correia, Vincenzi, Vanzella, Pinheiro, Lima, et al., 2022 [17] | .82 | .70 | .85 | N = 608 |
| Choi, 2021 [13] | .76 | .64 | | N = 104 |
| Average α/Assuming expected values | .78 | .66 | .82 | |
| Current study | N = 673 | N = 1235 | N = 862 | **N = 1235** |

## Cultural variability in musical ability

**Musical enculturation.**   The informal acquisition of musical knowledge through symbiosis with one's environment is termed "musical enculturation." This process significantly influences various aspects of music perception, including neural processing of syntactic complexity [33], music memory performance [34], narrative engagement with music [35], and melodic expectancy [36]. Crucially, the impact of cultural elements in one's lived environment on musical ability may precede formal musical training, imprinting a patterned effect on musical perceptions [14, 36]. Therefore, the ability of a music test to be applicable across diverse cultural systems becomes a crucial benchmark for its competence [10].

Notably, Chinese culture exhibits unique cultural heterogeneity, and its music culture encompasses a five-tone music theory system distinct from the Western music theory system. This, coupled with a unique aesthetic orientation, has given rise to a nuanced musical schema within the Chinese population. The Music Ear Test (MET), rooted in the Western tonal music theory system, may yield results influenced by cultural disparities, as suggested by Haumann et al. (2018) [14], particularly in the rhythmic sub-test when administered to non-Western listeners.

**China-specific native language differences.**   Numerous cognitive and developmental connections exist between music and language, including bidirectional transfer [13], the speech-to-song illusion transfer [37], shared cognitive neural bases [38], and the association of musical ability with phonological pitch discrimination [39]. Notably, speakers of tonal languages, such as Mandarin Chinese, exhibit distinct perceptions of music compared to non-native speakers of tonal languages [32, 40, 41]. Mandarin Chinese, being the most complex tonal language, presents unique associations between prosodic language speakers and music, necessitating independent study and differentiation from other native language samples [42]. This distinction is evident in correlation studies between MET and tonal languages like Cantonese [43] and Mandarin Chinese [32].

**Current landscape of music ability testing in China.**   China-specific cultural and native language differences, rooted in the theory of cultural variability, underscore the need for the translation and validation of musical ability tests in the Chinese context. Despite the myriad music ability tests available, there exists an underdevelopment of validated tests suitable for application in China. Initiatives to measure musical ability in Chinese populations began in the early 1990s [44, 45] and have continued with subsequent studies [46, 47]. While some Chinese researchers have endeavored to compile [48], adapt [47], and disseminate [49, 50] globally recognized tests, a comprehensive cross-cultural reliability and validity test for musical ability, openly accessible, remains elusive. This scarcity may be attributed to the absence of instruments for quantifying musical ability, the relatively low number of cross-cultural studies in musicology, and the prevalent categorization of studies among Chinese populations by self-reported musical skill levels [51].

The MET exhibits potential for effective cross-cultural distribution. Initially released in Danish by Wallentin et al. (2010) [23], it called for the inclusion of multiple language versions for cross-cultural validation. Subsequent studies explored its validity in groups of tonal language participants, noting a proclivity for high scores on melody subtests [18, 42]. Despite attempts by Chinese researchers to utilize the MET in their studies [32, 42, 52, 53], and possessing most indicators necessary for a qualifying test, it has not undergone specific Chinese adaptation or investigation for applicability in a Chinese sample, limiting its cross-cultural dissemination [54]. Building on existing research, this paper aims to develop a Chinese version of the MET and conduct a targeted assessment of its cross-cultural validity, scrutinizing its cultural idiosyncrasies. This endeavor aims to offer a valid indicator for quantifying the musical abilities of a Chinese sample.

### The present research

In light of the imperative need for the MET to serve effectively as a musical ability instrument, this study posits the following hypotheses:

**Factorial structure in cultural context.** Hypothesis 1: The factorial structure of the Chinese version of the MET is anticipated to demonstrate applicability within the specific framework of Chinese music/language culture.

Hypothesis 2: The factorial structure of the Chinese MET version is expected to exhibit a comparable functional pattern to versions utilized in diverse cultural backgrounds.

**Implications for musical ability.** Hypothesis 3: Results derived from a substantial sample through the MET questionnaire are hypothesized to provide clear indications of musical aptitude and expertise.

Hypothesis 4: The MET outcomes are expected to significantly differentiate populations situated in the middle spectrum, encompassing both musical aptitude and expertise.

## Materials

### General information questionnaire

This is a self-administered questionnaire for collecting basic information about the subjects. It consists of nine questions, takes approximately 02:00 min to complete, and contains self-reported information on the participants' age, gender, education/industry, level of musical education, mode of test completion, familiarity, and auditory status. Studies have shown that self-reports of musical skills can be reliably and objectively correlated with perceptual tests [55]. Therefore, the data collected through the questionnaire will be investigated in relation with the structure and content of the MET in the results section.

### Music Ear Test (MET)

The MET, devised and developed by Mikkel Wallentin and his research team in 2010, serves the purpose of identifying and measuring music competency [23]. Comprising 104 trials, each contributing to a total score of 104, respondents are tasked with discerning distinctions between two short music fragments within a specified time frame. The MET consists of two subtests: melody (52 trials) and rhythm (52 trials). In the melody subtest, focused on pitch and contour ability, the beat is set at 4/4, while the rhythm subtest, emphasizing rhythm and beat identification ability, operates at 100 beats per minute. Each subtest is accompanied by instructions containing two examples. A comprehensive description of the musical stimuli can be found in Swaminathan, Kragness, and Schellenberg [31].

The MET is publicly accessible, and all pertinent resources, including the answer sheet, correct answers, model, notation, and audio files (in both mp3 and WAV formats), were obtained from Mikkel Wallentin via email, with explicit permission from the original authors [23]. The right to adapt the original scale for use in the Chinese context was also granted.

The initial music test consisted of an audio assessment lasting 20 minutes and 30 seconds, encompassing both instructions and trials, and utilized a paper answer sheet for offline responses. In the context of this research, the test will undergo administration via a meticulously structured video presentation paired with a mobile answer sheet accessible both online and offline. The video will incorporate detailed explanations supported by subtitles, ensuring alignment with the original test's musical stimuli and time intervals. The duration of the new video format will maintain parity with the original audio duration of 20 minutes and 30 seconds.

The mobile answer sheet, fashioned after the model proposed by Correia et al., 2022 [17], will feature a progress bar at the bottom of each subtest page. This bar will indicate the

initiation and conclusion times for each subtest, displaying the total score—representing the number of correct questions—as a key metric for subsequent data analysis.

Both the General Information Questionnaire (GIQ) and the MET will be made publicly accessible as distinct sections, seamlessly integrated into a single link for convenient distribution.

## Method

### Participants

Convenience sampling techniques will be employed to recruit online participants, with an online questionnaire distributed among the Chinese community through social media platforms such as WeChat and E-mail. Simultaneously, the offline test will be administered centrally on site during face-to-face courses at Yanshan University. The inclusion criteria stipulate that participants must be (i) native Chinese speakers and (ii) healthy individuals without any degree of hearing impairment.

Prior to the commencement of the formal experiment scheduled for January 2024, a purposive sampling approach was utilized to select 82 healthy participants without hearing impairment for the pre-experiment. This selection was based on considerations such as age, gender, response method, and experience in music training. Participants will be categorized into four distinct groups, delineated by their acquisition of musical skills and level of music education: a) Musician: Participants with a music major background possessing more than three musical skills. b) Unskilled Musician: Participants with a music major background having less than three musical skills. c) Amateur: Individuals not specializing in music but possessing some musical skills. d) Non-Musician: Individuals without specialization in music and devoid of any musical skills. The demographics of the pre-experimental participants, along with the grouping of participants, are detailed in Table 2 (the complete sample profile will also be presented in the final paper).

### Translation

The translation of the Music Ear Test (MET) into Chinese followed standardized back-translation procedures [56] and underwent expert review. The primary focus of the translation process involved the MET's structured instructions and question items. Initially, two bilingual PhD members within our research group translated the MET from English to Chinese, generating the initial draft. This draft was subsequently forwarded to a proficient Chinese scholar fluent in both languages for back-translation into English. Following this, two bilingual psychology professors, well-versed in both languages, engaged in a comparative analysis between the back-translation and the original version. Through discussion, revisions were made to address any inconsistencies, ensuring alignment with the English version's length in the structured explanation. Additionally, for the structured explanation, more fluid and logically expressed phrases were incorporated, considering the characteristics of the Chinese language [57].

Following the translation process, the text underwent conversion into standard Mandarin Chinese audio using our intelligent audio-video generation software, Cut Image (剪映, 9.7.1). To enhance comprehension and rigorously control the test duration, the structured instruction was further transformed into a video format, featuring Chinese oral guidance and subtitles, maintaining a duration of 20 minutes and 30 seconds. The MET_CN full version is available at data available declarative statement.

### Data analysis

**Sample profile.** Following the recommendation by Ryan (2013), the sample size for this paper was calculated using the Normal Approximation method of the PASS software (PASS 15 Power Analysis and Sample Size Software, 2017, NCSS, Kaysville, Utah, USA, ncss.com/software/pass) [58]. To test the underlying hypotheses of this paper, we synthesized data based on the reliability of the MET (Cronbach's alpha, see Table 3) reported in multilingual samples from previous users. The mean values from these samples were averaged to establish baseline levels for reliability (overall reliability $\alpha = 0.82$, melody subtest $\alpha = 0.78$, rhythm subtest $\alpha = 0.66$). This baseline value is considered as P1 (alternative proportion) in effect size, aiming to mitigate potential errors associated with varying sample sizes, power, and effect size.

Furthermore, to ensure that the Chinese version of the MET meets the minimum reliability requirements, a relatively low value reported by Swaminathan et al.,2021 [31] was used as the null hypothesis (P0, null proportion). Assuming an alpha risk of 5%, beta risk of 10%, a dropout rate of 20%, and a difference (alternative proportion P1-P0) of 0.04 for the whole test, 0.05 for the melody subtest, and 0.04 for the rhythm subtest, we assert that a sample size of 862 (whole test), 673 (melody subtest), and 1235 (rhythm subtest) achieves 90% power to attain an acceptable level of reliability using a one-sided exact test with a significance level (alpha) of 0.005. For the current paper, a sample size of at least 1235 for each group will be considered adequate.

**Statistical analyses.** The Statistical Package for Social Sciences (SPSS, IBM Corp. Released 2016. IBM SPSS Statistics for Windows, Version 24.0. IBM Corp., Armonk, NY, USA) will be employed for various analyses in this study. Descriptive analyses, normality checks (utilizing the "+2/-2" and +10/-10 guidelines [59]), item analyses, exploratory factor analyses, correlation analyses, and reliability and validity tests will be conducted.

For internal consistency and folding reliability assessments of the scales, Cronbach's alpha coefficient and Gettleman's split-half reliability will be used. A value exceeding 0.70 will indicate good internal consistency and a high degree of split-half reliability. While the internal consistency of the MET has been well-documented in related studies (see Table 2), no information has been found regarding the formation of its dimensions. Therefore, common factor analyses will be conducted separately for the melodic and rhythmic subtests. The items will be clustered into factors for factorial structure analysis.

Additionally, multiple regression analyses will be performed, incorporating formative indicators such as age, gender, major, mode of answering (on/off line), and level of music education. This analysis aims to elucidate score variations and differentiate the results among the four cohorts of Chinese samples.

To assess the reliability of the test over time, 10% of the subjects from the sample will be retested after 4 weeks, and Pearson correlation analysis will be employed for the retest reliability evaluation. Factor analysis will be employed to assess the structural validity of the scale.

Confirmatory factor analysis (CFA) and measurement equivalence tests were conducted using IBM SPSS AMOS version 24.0.0. The goodness-of-fit criteria for the models included $\chi 2/df < 5$, Tucker-Lewis index (TLI) and comparative fit index (CFI) both $> 0.95$, and root mean square error of approximation (RMSEA) $< 0.08$ [60].

Comparisons will be made with data obtained from Correia, et al. (2022) [17] and Swaminathan et al. (2021) [31] in culturally diverse populations. Driven by theories about the measurement of musical ability, this study aims to determine the best model to represent the data from the Chinese sample based on these statistics and will also examine the best model for the Chinese translated version. This process ensures that the translated Chinese version of the MET maintains item equivalence and the measurement equivalence of the Universal model in the Chinese population.

Multigroup CFAs were employed to examine the measurement invariance of the Chinese sample. Specifically, the change in CFI ($\Delta$CFI) and the change in RMSEA ($\Delta$RMSEA) were used as the main metrics. An acceptable measurement equivalence model is indicated when $\Delta$CFI < 0.01 and $\Delta$RMSEA < 0.015 [54]. The test level $\alpha$ = 0.05, with P<0.05 being a statistically significant difference. Convergent validity of MET will also be a potentially important indicator for this validation.

Structural equation modeling (SEM) will be employed to examine the predictive validity of the MET. Specifically, a model will be set up to estimate the structure of the MET (i.e., melody and rhythm), set as a predictor of musical aptitude. The second model will set the structure of the MET as a predictor of musical expertise. The adequacy of the models will be determined using the same criteria as for the assessment factor model. The mediating effect of music acquisition will be measured in both models using maximum likelihood. Significant indirect effects are inferred when zero is not included within the upper and lower 95% confidence intervals.

## Procedure

The study protocol has received approval from the Research Ethics Board at Qinhuangdao First Hospital (2023G010-2). The MET is a publicly available test, and all related resources, including the answer sheet, correct answers, model, notation, and audio (in mp3 and WAV format), were obtained from Mikkel Wallentin via email, with the permission of the original authors [23] and the right to update the original scale in Chinese.

Data collection will be executed through the local Chinese questionnaire platform "Wenjuan.com" (问卷网). Participants will receive information about the study's meaning, purpose, and details through a letter before starting the test. Upon reading the letter and initiating the test, participants will be deemed to have provided consent for the use of the information they provide in the test for this study. Participation is entirely voluntary, and participants retain the right to terminate the test at any time. The online participant sample will access the complete questionnaire, including the video and answer sheet, through a QR code scan or a web page link. The offline participant sample will watch the video and listen to the audio through a standardized speaker on a large screen at the front of the classroom. It is crucial to note that both online and offline participants will answer the questions via the website on their mobile devices and will select their answer method in the first section. Finally, an expected 50 subjects will be selected for retesting through random sampling within two weeks after the questionnaire recall.

During data collection and pre-processing, questionnaires meeting any of the following criteria will be considered invalid: (i) participants who self-report a hearing impairment in the first part, and (ii) participants who answer the questions in less than 20 minutes and 30 seconds or take much longer.

## Pre-testing and future result

**Sinicization results.** The MET underwent translation into Chinese following standardized back-translation procedures. While respecting the original meaning of the scale, adjustments were made to expressions that did not conform to the rules of the Chinese language. Subsequently, through a cultural debugging procedure, in-depth interviews were conducted with pre-test participants to understand their interpretation of the test items. The Chinese version was then modified based on their feedback and answers, while retaining the original structure of the final scale. This Chinese version of the Music Ear Test was utilized as the material completed by the pre-experimental subjects.

**Table 4. The MET index scores situation (pre-test data n = 82).**

| Dimensionality | GROUP | MEAN | SD | MIN | MAX | Range | Skewness | Kurtosis |
|---|---|---|---|---|---|---|---|---|
| Met Score | Whole | 67.3 | 12.3 | 41 | 91 | 50 | -0.83 | -1.10 |
| | Musician | 75.5 | 20.8 | 45 | 91 | 46 | -1.72 | 3.08 |
| | Amateur | 67.1 | 13.0 | 52 | 83 | 31 | -0.14 | -2.15 |
| | Non-musician | 64.4 | 10.6 | 41 | 87 | 46 | 0.03 | -0.84 |
| | Unskilled musician | 79.3 | 9.8 | 50 | 90 | 40 | -2.74 | 8.81 |
| Melody Subtest | Whole | 34.21 | 6.8 | 12 | 47 | 32 | -0.2 | -0.52 |
| | Musician | 38.5 | 11.7 | 21 | 46 | 25 | -1.93 | 3.79 |
| | Amateur | 33.9 | 6.4 | 26 | 42 | 16 | -0.22 | -1.67 |
| | Non-musician | 32.7 | 5.8 | 15 | 46 | 31 | -0.22 | 0.14 |
| | Unskilled musician | 40.6 | 5.9 | 24 | 47 | 23 | -2.11 | 5.87 |
| Rhythm Subtest | Whole | 32.68 | 6.23 | 21 | 44 | 23 | -0.04 | -1.14 |
| | Musician | 36.5 | 8.3 | 25 | 44 | 19 | -1.16 | 0.97 |
| | Amateur | 32.6 | 6.7 | 26 | 40 | 14 | 0.09 | -2.50 |
| | Non-musician | 31.4 | 5.7 | 21 | 42 | 21 | 0.09 | -0.89 |
| | Unskilled musician | 37.9 | 4.9 | 26 | 42 | 16 | -1.42 | 1.91 |

**Scale score statistics.** The scale score statistics for the pre-experimental sample are presented in advance, as shown in Table 4. Overall, the data for the different subject groups exhibited differences. With the exception of the Unskilled musician subgroup, the overall sample demonstrated a normal distribution (Skewness within ±2 and Kurtosis within ±10), suggesting that the data can be analyzed using Maximum Likelihood for further Factorial Structure analysis.

Additionally, a preliminary reliability analysis was conducted, revealing strong validity for the Chinese version of the MET (response rate = 100%, Cronbach's α = 0.858). This aligns closely with the average validity data (Cronbach's α = 0.82). Similarly, robust performance was observed in both sub-tests (Melody subtest α = 0.772, Rhythm subtest α = 0.778). Preliminary data analysis has indicated preliminary support for the hypotheses presented in this article.

**Factorical structure.** Assuming normal distribution of the dataset, this section will elucidate the outcomes of confirmatory factor analysis (CFA) involving multiple competing models. Following scrutiny of parameter estimates and modification indices, adjustments may be made for items displaying weak loadings in the dataset.

Factor loadings and factor correlations for MET across the four distinct samples representing China will be presented in Table 5. The reliability of these presentations will be comparable to similar data obtained in other samples. It is important to note that the structure of the data presentation in Table 6 assumes the validity of the two-factor model and may be adjusted accordingly based on the results of this section.

**Measurement invariance.** Model fit for the CFA configural models will be established across the four groups. Scalar invariance of the MET results and verification of the MET results for different musical abilities can only be established through model comparisons.

**Table 5. Fit indices for confirmatory factor analysis models of the Music Ear Test (MET).**

| Model | χ2 | df | CFI | TLI | RMSEA (90% CI) | SRMR |
|---|---|---|---|---|---|---|
| Whole sample CFAs for 104-item scale one-factor model | | | | | | |
| Two-factor model (melody-rhythm) | | | | | | |

Note: χ2 = Chi-square; df = Degrees of freedom; CFI = Comparative fit index; TLI = Tucker Lewis fit index; RMSEA = Root mean square error of approximation; SRMR = Standardised root mean square of residuals. 1 = no convergence.

**Table 6. Means, standard deviations, reliability coefficients, average variance extracted, standardised factor loadings, item residual variances, and factor correlations for the Chinese versions of the Music Ear Test (MET).**

| Factors/items | | GROUP | | | | | | | | | | | | | | | | |
|---|---|---|---|---|---|---|---|---|---|---|---|---|---|---|---|---|---|---|
| | | Musician | | | | Amateur | | | | Non-musician | | | | Unskilled musician | | | |
| | | M | SD | λ | δ | M | SD | λ | δ | M | SD | λ | δ | M | SD | λ | δ |
| Factor 1 | 1 | - | - | | | | | | | | | | | | | | |
| | 2 | | | | | | | | | | | | | | | | |
| | 3 | | - | - | | | | | | | | | | | | | |
| | 4 | | | | | | | | | | | | | | | | |
| | 5 | | | | | | | | | | | | | | | | |
| Factor2 | 6 | | | | | | | | | | | | | | | | |
| | 7 | | | | | | | | | | | | | | | | |
| | 8 | | | | | | | | | | | | | | | | |
| | 9 | | | | | | | | | | | | | | | | |
| | 10 | | | | | | | | | | | | | | | | |
| φ | | F1 | | F2 | | F1 | | F2 | | F1 | | F2 | | F1 | | F2 | |
| F1 | | | | | | | | | | | | | | | | | |
| F2 | | | | | | | | | | | | | | | | | |
| α | | | | | | | | | | | | | | | | | |
| ω | | | | | | | | | | | | | | | | | |
| ρ | | | | | | | | | | | | | | | | | |
| AVE | | | | | | | | | | | | | | | | | |

Note. f1 = Melody factor, f2 = Rhythm factor, λ = Standardised factor loading, δ = Standardised residual variance, φ = Factor correlations, α = Cronbach's alpha values, ω = McDonald's omega values, ρ = Composite reliability coefficient; AVE = Average variance extracted.

**Convergent validity.** The data presented in this section will identify the most relevant patterns of predictive relationships for musical ability among several models through potential factor correlations.

**Predictive validity.** The results of the data in this section have implications for the validity of the MET construct as a predictor of musical ability or musical skill. This component will determine the pattern and level of significance of the impact of musical ability or musical skills by testing the direct and indirect intervention pathways of music acquisition levels on MET outcomes.

## Limitation

Although we have endeavored to consider the relevant caveats, there are unfinished elements to this study. Firstly, while there is research suggesting that self-reporting can be a valid indication of the level of musical acquisition [55], a more comprehensive approach to the assessment of musical training, such as using other well-established and specialized scales, may be more effective. Secondly, the different aspects involved in the definition of musical ability should be more widely incorporated into multidisciplinary studies, including those with physiological data on musical cognitive ability, to obtain more comprehensive results [19].

## Supporting information

**S1 File. Timeline.**
(DOCX)

## Author Contributions

**Conceptualization:** Xiaoyu Wang.

**Data curation:** Xiaoyu Wang, Shilin Liu.

**Formal analysis:** Xiaoyu Wang, Shilin Liu.

**Funding acquisition:** Shidan Wang.

**Investigation:** Dan Yang, Yintong Liu.

**Methodology:** Xiaoyu Wang, Dan Yang, Meihui Li.

**Project administration:** Xiaoyu Wang.

**Resources:** Xiaoyu Wang, Xiubo Ren, Shidan Wang.

**Software:** Dan Yang.

**Supervision:** Qiujian Xu.

**Validation:** Shidan Wang.

**Visualization:** Mingyi Yang.

**Writing – original draft:** Xiaoyu Wang, Xiubo Ren, Qiujian Xu.

**Writing – review & editing:** Xiaoyu Wang, Qiujian Xu.

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
