## [Decision Letter · Decision Letter 0]

30 Oct 2023

PONE-D-23-24643Validation of the Chinese version of the Music Ear Test on a Large Chinese SamplePLOS ONE

Dear Dr. Xu,

Thank you for submitting your manuscript to PLOS ONE. After careful consideration, we feel that it has merit but does not fully meet PLOS ONE’s publication criteria as it currently stands. Therefore, we invite you to submit a revised version of the manuscript that addresses the points raised during the review process. Please submit your revised manuscript by Dec 14 2023 11:59PM. If you will need more time than this to complete your revisions, please reply to this message or contact the journal office at plosone@plos.org. Please include the following items when submitting your revised manuscript:A rebuttal letter that responds to each point raised by the academic editor and reviewer(s). You should upload this letter as a separate file labeled 'Response to Reviewers'.A marked-up copy of your manuscript that highlights changes made to the original version. You should upload this as a separate file labeled 'Revised Manuscript with Track Changes'.An unmarked version of your revised paper without tracked changes. You should upload this as a separate file labeled 'Manuscript'.

We look forward to receiving your revised manuscript.

Kind regards,

Hamed Ahmadinia

Academic Editor

PLOS ONE

Journal Requirements:

2. In your cover letter, please confirm that the research you have described in your manuscript, including participant recruitment, data collection, modification, or processing, has not started and will not start until after your paper has been accepted to the journal (assuming data need to be collected or participants recruited specifically for your study). In order to proceed with your submission, you must provide confirmation.

"The funders did not and will not have a role in study design, data collection and analysis, decision to publish, or preparation of the manuscript." 

Additional Editor Comments:

Dear Authors,

Thank you for submitting your manuscript to PLOS ONE. I appreciate the effort you have put into this work and the depth of your study. After a thorough review by multiple experts, I have decided to request major revisions before considering your work for publication in our journal.

The reviewers and I have identified several critical areas that require attention. Below is a synthesis of the most pressing concerns, which we hope you address in your revised manuscript:

Conceptualization of Musical Ability: A discrepancy has been noted between the musical ability definition you've chosen and the intent of the original Music Ear Test (MET). The MET was designed primarily for "musical expertise." If the objective is to create a MET variant for gauging musical potential in children before formal training, the MET might not be the optimal choice.

Translation Contribution and Scale Validation: Given the MET's almost non-linguistic design, the significance of translating such a test becomes questionable. Given this, justifying the need for large-scale validation for what seems like a modest translation is crucial.

Rationale for Power Analysis: There's ambiguity in the values associated with the null and alternative hypotheses in your power analysis. A clearer rationale aligning the study's goal with the chosen values for this analysis is imperative.

Data Accessibility: The data access link provided was problematic. The data used in PLOS ONE publications must be easily accessible to reviewers and readers. Kindly ensure compliance with the PLOS Data Policy.

Narrative Enhancement and References: The manuscript's narrative requires a richer touch, veering away from its current technical guide-like nature. Additionally, an update to your references is advisable, emphasising recent papers (from the last five years) from reputable journals.

Clarification on Analysis and Proofreading: Further clarity is sought in the sections detailing power analysis and data analysis. Specific tests and distributions used in the power analysis should be outlined, and the data analysis plan should be more descriptive. Typos and grammatical errors need rectification for manuscript quality and professionalism.

Discussion and Conclusion Sections: The manuscript appears to lack depth in the discussion and conclusion sections. These sections are pivotal for positioning the research in a broader context and addressing its implications and significance. It is essential to expand upon these areas to provide comprehensive insights into the study's outcomes and potential impact.

Besides these major points, the reviewers provided other recommendations and comments that should be addressed in your revised manuscript. Taking these comments seriously and addressing them comprehensively is strongly advised.

Upon completion of the revisions, kindly resubmit your manuscript along with a detailed point-by-point response to each comment from the reviewers.

Thank you for choosing PLOS One for your research. We anticipate receiving your revised manuscript.

Sincerely,

Academic Editor,

PLOS ONE

Reviewers' comments:

Reviewer's Responses to Questions

**Comments to the Author**

1. Does the manuscript provide a valid rationale for the proposed study, with clearly identified and justified research questions?

Reviewer #1: No

Reviewer #2: Yes

Reviewer #3: Partly

2. Is the protocol technically sound and planned in a manner that will lead to a meaningful outcome and allow testing the stated hypotheses?

Reviewer #1: No

Reviewer #2: Yes

Reviewer #3: No

3. Is the methodology feasible and described in sufficient detail to allow the work to be replicable?

Reviewer #1: No

Reviewer #2: Yes

Reviewer #3: No

4. Have the authors described where all data underlying the findings will be made available when the study is complete?

Reviewer #1: No

Reviewer #2: Yes

Reviewer #3: No

5. Is the manuscript presented in an intelligible fashion and written in standard English?

Reviewer #1: Yes

Reviewer #2: Yes

Reviewer #3: Yes

6. Review Comments to the Author

You may also provide optional suggestions and comments to authors that they might find helpful in planning their study.

Reviewer #1: The author seems to be planning to revalidate a questionnaire about music in a cross-cultural context. My concern is that I fail to understand the significance of doing so. Especially when you require a sample size of thousands, merely to validate a questionnaire that has already been peer-reviewed and in use for over a decade. In other words, even if the reliability and validity of this scale are reaffirmed through your paper, what specific implications does it have for practical management or theoretical development? I hope the author provides a more comprehensive explanation and clarification on this point. Additionally, the narrative aspect of this manuscript needs enhancement. The author does not effectively tell a story; the current content reads more like a dry textbook or research guide. Regarding the references, I suggest updating all citations, that means, you should only use the paper from the past 5 years. Also, consider including only papers published in reputable journals as your references.

Reviewer #2: This manuscript describes a research protocol to develop and validate a Chinese version of the Music Ear Test.

The manuscript is generally acceptable and provides all the necessary details of the protocol. However, the authors neglected to include a section on this study's limitations and potential pitfalls. If PLOS ONE requires such a section in protocol papers, it must be returned for minor revision to include a section on limitations. Otherwise, this paper is acceptable.

Reviewer #3: WANG and colleagues propose a large-scale study aimed at validating the Chinese adaptation of the Musical Ear Test, focusing on Mandarian speakers with diverse levels of musical expertise. While the preregistration report presents a solid foundation, there are areas that could benefit from further refinement. I will now elaborate on these points.

The page numbers indicated as [Pxx] are derived from the merged PDF, including meta information and the manuscript file provided to the reviewer. To streamline the editorial process, please include continuous line numbers within the manuscript.

# Major points #

[P09] The conceptualization of musical ability in this study appears to deviate from that of the original test.

The authors have adopted a definition of musical ability characterized as the "potential for learning music" (Shuter-Dyson, 1999), particularly before formal training, a concept akin to "musical aptitude (or talent)" (Gordon, 2007). However, the Musical Ear Test (MET; Wallentin et al., 2010) is primarily designed to assess "musical expertise." It has demonstrated the ability to distinguish between professional musicians, amateur musicians, and non-musicians, and it correlates with the amount of musical practice. Consequently, it is clear that the perceptual skill assessed by the MET is indicative not only of potential before formal training but also of expertise acquired through training, particularly in adults. Therefore, given the authors aim to establish a Mandarin version of a behavioral test that measures musical potential in young children before formal training, it is inconsistent to use a behavioral test originally developed for adults such as MET.

[P16] The contriution of a translation appears to be quite trivial.

Is a large-scale study truly necessary to validate a translated version of a test that is almost non-linguistic in nature? The Musical Ear Test (MET) is designed in a straightforward manner, with the primary task being to listen to two phrases and determine whether they are identical or not. Therefore, the main contribution of this study would be the translation of this instruction: "Listen to two phrases and tell whether they are identical or not" (听两个短语，并判断它们是否相同 [translated by deepl.com]). This impression is in part due to the unavailbility of the translated materials (see below) but also due to the nature of the perceptual test.

[P17] The rationale for the power analysis is unclear.

The authors averaged Cronbach's alpha values from three of the four studies with higher values and considered this average as the expected value under the alternative hypothesis. They also took the lowest value, as reported in Swaminathan et al. (2021), as the expected value under the null hypothesis. The reviewer does not understand why these values are associated with the null and alternative hypotheses of the current study. It raises questions about whether the primary goal of this large-scale study is simply to obtain a sample with a higher Cronbach's alpha than one of the previous studies, which would appear puzzling and largely irrelevant to the objectives the authors presented.

It seems more relevant to frame the research question as whether the reliability of the Mandarin version in a Chinese population is similar to that of the Danish version in a Danish population, which was the basis for the original study.

[P19] Data access was problematic.

The reviewer encountered difficulties when attempting to access the data from the provided link (https://www.wenjuan.pub/s/UZBZJv98ms/). This issue appears to stem from the website's exclusive support for the Chinese language. Using Google Translate, the message displayed on the page was translated to "Collection of this review has been suspended" (该测评已暂停收集) as of October 22, 2023. Despite attempting various options on the webpage, the reviewer was unable to download any files and was consistently redirected to advertisement pages. Hence, the reviewer was unable to obtain any details regarding the authors' translation of the MET. That is, in contrast to their data availability statement, the data is not accessible and prepared for evaluation.

[P21] Data availability is unclear.

The metadata table indicates that the authors have declared, "Yes - any pilot data reported in this submission are fully available, and data collected during the study will be made fully available without restriction upon study completion." However, this protocol mentions a "pre-testing" pilot study involving 20 participants, and yet the data from this pilot study is not available in the submission. This raises concerns about whether this submission is in compliance with the PLOS Data Policy.

Furthermore, it is crucial to include specific details about how the primary dataset will be made available, which should encompass aspects such as obtaining informed consent for data publication and utilizing internationally accessible platforms like the Open Science Framework.

# Minor points #

[P17] The description of the power analysis is unclear, particularly in regard to the test or distribution used to compare Cronbach's alphas. It would be helpful to provide more details on the statistical methods and the specific tests or distributions used in the power analysis. Clarity in describing the statistical procedures and how the power analysis was conducted will enhance the transparency of your research and help readers understand the basis for your sample size determination.

[P19] The description of the data analysis is rather vague. It's not clear which specific tests are being performed to test what relationships or differences, what the expected outcomes are, or what planned contrasts will be used. It would be beneficial to provide more details and clarity regarding the specific statistical tests and analyses that will be conducted, along with the expected outcomes and any planned contrasts or comparisons that are relevant to the study. This will help readers and reviewers better understand the analysis plan and the research objectives.

- It's important to address typos and grammatical errors in your manuscript to ensure clarity and professionalism. Some specific examples are "[P19] Date Analyses" and "[P21] 1830 years." It's advisable to thoroughly proofread and copy-edit the manuscript before submission to ensure that such issues are corrected. This will help enhance the overall quality of the paper.

7. PLOS authors have the option to publish the peer review history of their article (what does this mean?). If published, this will include your full peer review and any attached files.

Reviewer #1: **Yes: **Chao Gu

Reviewer #2: **Yes: **Dr. Ali M. AL-Asadi

Reviewer #3: No

---

## [Author Response · Author response to Decision Letter 0]

18 Dec 2023

Comments from the Editors and Reviewers and Reply from the authors

PONE-D-23-24643

Validation of the Chinese version of the Music Ear Test on a Large Chinese Sample

PLOS ONE

Conceptualization of Musical Ability: A discrepancy has been noted between the musical ability definition you've chosen and the intent of the original Music Ear Test (MET). The MET was designed primarily for "musical expertise." If the objective is to create a MET variant for gauging musical potential in children before formal training, the MET might not be the optimal choice.

Re: We sincerely apologize for the confusion caused by the wording in our manuscript. Indeed, the original manuscript stated that "Musical ability refers to the potential for learning music." However, our original intention was to emphasize the dual directionality within the Music Ear Test (MET), suggesting that it may encompass both music ability and music expertise.

Moreover, the conceptualization of Musical Ability has been a contentious issue, especially concerning its application in MET. In our review of existing studies, we found that researchers have employed MET with different emphases, including music expertise, music aptitude, and auditory discrimination ability. In fact, elucidating the directionality of MET results through a refined categorization of the musical subject group is one of the crucial objectives of our study.

To address this issue, we have made substantial revisions to the group related to musical ability, including the addition of a dedicated "Definition of Musical Ability" section to clarify this matter. Additionally, we have incorporated this issue as a primary research focus in our hypotheses, data analysis, and conclusions.

Translation Contribution and Scale Validation: Given the MET's almost non-linguistic design, the significance of translating such a test becomes questionable. Given this, justifying the need for large-scale validation for what seems like a modest translation is crucial.

Re: The validation of MET in a Chinese context comprises two main aspects: questionnaire translation and the factorial structure in a cultural context. The primary emphasis during the translation process was on the MET's structured instructions and question items. This Chinese adaptation ensures the effectiveness of the questionnaire process and facilitates its widespread use in China. The examination of the factorial structure in a cultural context is a crucial aspect of large-scale validation.

On one hand, China-specific cultural and native language differences, rooted in the theory of cultural variability, highlight the necessity for translating and validating musical ability tests in the Chinese context. Concerns have been raised within the academic community about the applicability of MET to non-Western audiences (Haumann et al., 2018). Participants from tonal languages also exhibited trends in the rhythm subtest that differed from the average performance.

On the other hand, the current lack of effective tools to quantify musical ability in relevant Chinese domains severely limits the development of interdisciplinary research in musicology. In summary, we believe that the translation and validation of MET on a large Chinese sample are necessary and significant. To address potential similar concerns, we have reorganized the logical presentation of this viewpoint; for further details, please refer to the "Cultural Variability in Musical Ability" section of our article.

Rationale for Power Analysis: There's ambiguity in the values associated with the null and alternative hypotheses in your power analysis. A clearer rationale aligning the study's goal with the chosen values for this analysis is imperative.

The authors have given careful consideration to your suggestions.

Firstly, the authors extensively collected reliability data reported in existing studies. In the end, only four studies in Table 2 reported such data, while the average was derived from the overall MET reliability. Your misunderstanding stemmed from the unclear expression in our presentation, and we have since revised the corresponding statements in the hope of addressing this issue.

Secondly, while Wallentin et al.'s (2010) study was conducted at a Danish university, and the authors stated that all participants had Danish as their first language, demographic information in the article did not provide an accurate description of nationality. Considering the mention of providing an English version of MET and the potential inclusion of foreign participants, we could not conclusively determine if the reported reliability came from the Danish version.

Thirdly, Wallentin et al.'s (2010) results may have some potential limitations. The reported reliability is based on a relatively small sample size (n=60), without power calculation and effect size reporting, which may limit the generalizability of the sample.

Considering these concerns, we chose to use the average reliability data from all existing studies as the P1 (alternative proportion) assumption to test the fundamental hypothesis of our study, "The validity of MET on Chinese samples can achieve reliability values reported in MET studies based on multilingual samples." This approach aims to mitigate potential errors introduced by variations in sample size, power, and effect size.

Correspondingly, to address potential reader concerns, we have updated the wording in the Data Analysis and Sample Profile sections of the article.

These are our thoughts on the matter, and we believe your input is crucial. If you feel that these considerations are unnecessary, we are willing to make adjustments to the calculation of sample size at any time.

Data Accessibility: The data access link provided was problematic. The data used in PLOS ONE publications must be easily accessible to reviewers and readers. Kindly ensure compliance with the PLOS Data Policy.

The authors have now uploaded all relevant data to the Open Science Framework(https://osf.io/kxey3/?view_only=e97871f8520347279c807022e29e381d) and updated the contact information within the manuscript.

Narrative Enhancement and References: The manuscript's narrative requires a richer touch, veering away from its current technical guide-like nature. Additionally, an update to your references is advisable, emphasising recent papers (from the last five years) from reputable journals.

In response to this issue, we have refined the language and expression throughout the entire manuscript. The text has also been proofread by native speakers for further professionalism. Additionally, we have made efforts to update the list of references, eliminating outdated and unnecessary citations. 

Clarification on Analysis and Proofreading: Further clarity is sought in the sections detailing power analysis and data analysis. Specific tests and distributions used in the power analysis should be outlined, and the data analysis plan should be more descriptive. Typos and grammatical errors need rectification for manuscript quality and professionalism.

We sincerely apologize for any inconvenience caused by this issue. To address it, we have first updated the details of the power analysis. We have completely rewritten and enriched the steps of data analysis, undergoing thorough and multiple proofreading processes throughout the entire manuscript. We hope that these measures effectively resolve the identified issues in the manuscript. 

Discussion and Conclusion Sections: The manuscript appears to lack depth in the discussion and conclusion sections. These sections are pivotal for positioning the research in a broader context and addressing its implications and significance. It is essential to expand upon these areas to provide comprehensive insights into the study's outcomes and potential impact.

In response to the comprehensive new results section, we have expanded on the significance and impact of the study. However, due to the writing requirements of the registration report, we were unable to validate the hypotheses on a sufficiently large sample size. As a result, most conclusions remain in the hypothetical stage. We are eagerly looking forward to the opportunity to advance the progress of the manuscript.

Reviewers' comments:

Reviewer #1: The author seems to be planning to revalidate a questionnaire about music in a cross-cultural context. My concern is that I fail to understand the significance of doing so. Especially when you require a sample size of thousands, merely to validate a questionnaire that has already been peer-reviewed and in use for over a decade. In other words, even if the reliability and validity of this scale are reaffirmed through your paper, what specific implications does it have for practical management or theoretical development? I hope the author provides a more comprehensive explanation and clarification on this point. Additionally, the narrative aspect of this manuscript needs enhancement. The author does not effectively tell a story; the current content reads more like a dry textbook or research guide. Regarding the references, I suggest updating all citations, that means, you should only use the paper from the past 5 years. Also, consider including only papers published in reputable journals as your references.

Re:

Dear review#1: 

Thank you for providing invaluable suggestions for this manuscript; your insights are crucial for enhancing its quality. 

Regarding the articulation of the research significance, we acknowledge previous shortcomings. The significance of our study can be delineated into two major aspects: Factorial Structure in Cultural Context and Implications for Musical ability. This study aims to assess the effectiveness of the Music Ear Test (MET) as a tool for gauging musical ability. The hypotheses posit that the factorial structure of the Chinese MET version will align with the specific nuances of Chinese music and language culture, as well as demonstrate a functional pattern comparable to versions used in diverse cultural contexts. Additionally, MET results from a substantial sample are expected to provide clear indications of musical aptitude and expertise, while also significantly differentiating populations within the middle spectrum encompassing both facets of musical ability.

To address potential similar concerns, we have reorganized the logical presentation of this viewpoint.

In addition, we have refined the language and expression throughout the entire manuscript. The text has also been proofread by native speakers for further professionalism. Additionally, we have made efforts to update the list of references, eliminating outdated and unnecessary citations. 

We look forward to any opportunity for improving the quality of the manuscript and eagerly await your response.

Reviewer #2: This manuscript describes a research protocol to develop and validate a Chinese version of the Music Ear Test.

The manuscript is generally acceptable and provides all the necessary details of the protocol. However, the authors neglected to include a section on this study's limitations and potential pitfalls. If PLOS ONE requires such a section in protocol papers, it must be returned for minor revision to include a section on limitations. Otherwise, this paper is acceptable.

Re:

Dear review#2: 

We sincerely appreciate your encouragement and support for this manuscript.

In response to your feedback, we have added a "Limitations" section.

Best wishes.

Reviewer #3: WANG and colleagues propose a large-scale study aimed at validating the Chinese adaptation of the Musical Ear Test, focusing on Mandarian speakers with diverse levels of musical expertise. While the preregistration report presents a solid foundation, there are areas that could benefit from further refinement. I will now elaborate on these points.

The page numbers indicated as [Pxx] are derived from the merged PDF, including meta information and the manuscript file provided to the reviewer. To streamline the editorial process, please include continuous line numbers within the manuscript.

Dear review：

Thank you for your support, and your valuable feedback has played a significant role in improving the quality of this manuscript. Below is a point-by-point response from the authors addressing your review comments.

Best wishes;

# Major points #

[P09] The conceptualization of musical ability in this study appears to deviate from that of the original test.

The authors have adopted a definition of musical ability characterized as the "potential for learning music" (Shuter-Dyson, 1999), particularly before formal training, a concept akin to "musical aptitude (or talent)" (Gordon, 2007). However, the Musical Ear Test (MET; Wallentin et al., 2010) is primarily designed to assess "musical expertise." It has demonstrated the ability to distinguish between professional musicians, amateur musicians, and non-musicians, and it correlates with the amount of musical practice. Consequently, it is clear that the perceptual skill assessed by the MET is indicative not only of potential before formal training but also of expertise acquired through training, particularly in adults. Therefore, given the authors aim to establish a Mandarin version of a behavioral test that measures musical potential in young children before formal training, it is inconsistent to use a behavioral test originally developed for adults such as MET.

Re: We sincerely apologize for the confusion caused by the wording in our manuscript. Indeed, the original manuscript stated that "Musical ability refers to the potential for learning music." However, our original intention was to emphasize the dual directionality within the Music Ear Test (MET), suggesting that it may encompass both music ability and music expertise.

Moreover, the conceptualization of Musical Ability has been a contentious issue, especially concerning its application in MET. In our review of existing studies, we found that researchers have employed MET with different emphases, including music expertise, music aptitude, and auditory discrimination ability. In fact, elucidating the directionality of MET results through a refined categorization of the musical subject group is one of the crucial objectives of our study.

To address this issue, we have made substantial revisions to the group related to musical ability, including the addition of a dedicated "Definition of Musical Ability" section to clarify this matter. Additionally, we have incorporated this issue as a primary research focus in our hypotheses, data analysis, and conclusions.

[P16] The contriution of a translation appears to be quite trivial.

Is a large-scale study truly necessary to validate a translated version of a test that is almost non-linguistic in nature? The Musical Ear Test (MET) is designed in a straightforward manner, with the primary task being to listen to two phrases and determine whether they are identical or not. Therefore, the main contribution of this study would be the translation of this instruction: "Listen to two phrases and tell whether they are identical or not" (听两个短语，并判断它们是否相同 [translated by deepl.com]). This impression is in part due to the unavailbility of the translated materials (see below) but also due to the nature of the perceptual test.

RE: The validation of MET in a Chinese context comprises two main aspects: questionnaire translation and the factorial structure in a cultural context. In the article, we acknowledge that our mention of only the translation of questionnaire items was an oversight on our part. We have now updated the relevant descriptions to address this omission. The primary emphasis during the translation process was on the MET's structured instructions and question items. This Chinese adaptation ensures the effectiveness of the questionnaire process and facilitates its widespread use in China. The examination of the factorial structure in a cultural context is a crucial aspect of large-scale validation.

On one hand, China-specific cultural and native language differences, rooted in the theory of cultural variability, highlight the necessity for translating and validating musical ability tests in the Chinese context. Concerns have been raised within the academic community about the applicability of MET to non-Western audiences (Haumann et al., 2018). Participants from tonal languages also exhibited trends in the rhythm subtest that differed from the average performance.

On the other hand, the current lack of effective tools to quantify musical ability in relevant Chinese domains severely limits the development of interdisciplinary research in musicology. In summary, we believe that the translation and validation of MET on a large Chinese sample are necessary and significant. To address potential similar concerns, we have reorganized the logical presentation of this viewpoint; for further details, please refer to the "Cultural Variability in Musical Ability" section of our article.

Furthermore, I sincerely apologize for the broken links. To address this, I have uploaded the updated versions of the video and translated material text to OSF for your reference. I hope this resolves any concerns you may have.

[P17] The rationale for the power analysis is unclear.

The authors averaged Cronbach's alpha values from three of the four studies with higher values and considered this average as the expected value under the alternative hypothesis. They also took the lowest value, as reported in Swaminathan et al. (2021), as the expected value under the null hypothesis. The reviewer does not understand why these values are associated with the null and alternative hypotheses of the current study. It raises questions about whether the primary goal of this large-scale study is simply to obtain a sample with a higher Cronbach's alpha than one of the previous studies, which would appear puzzling and largely irrelevant to the objectives the authors presented.

It seems more relevant to frame the research question as whether the reliability of the Mandarin version in a Chinese population is similar to that of the Danish version in a Danish population, which was the basis for the original study.

Re: 

The authors have given careful consideration to your suggestions.

Firstly, the authors extensively collected reliability data reported in existing studies. In the end, only four studies in Table 2 reported such data, while the average was derived from the overall MET reliability. Your misunderstanding stemmed from the unclear expression in our presentation, and we have since revised the corresponding statements in the hope of addressing this issue.

Secondly, while Wallentin et al.'s (2010) study was conducted at a Danish university, and the authors stated that all participants had Danish as their first language, demographic information in the article did not provide an accurate description of nationality. Considering the mention of providing an English version of MET and the potential inclusion of foreign participants, we could not conclusively determine if the reported reliability came from the Danish version.

Thirdly, Wallentin et al.'s (2010) results may have some potential limitations. The reported reliability is based on a relatively small sample size (n=60), without power calculation and effect size reporting, which may limit the generalizability of the sample.

Considering these concerns, we chose to use the average reliability data from all existing studies as the P1 (alternative proportion) assumption to test the fundamental hypothesis of our study, "The validity of MET on Chinese samples can achieve reliability values reported in MET studies based on multilingual samples." This approach aims to mitigate potential errors introduced by variations in sample size, power, and effect size.

Correspondingly, to address potential reader concerns, we have updated the wording in the Data Analysis and Sample Profile sections of the article.

These are our thoughts on the matter, and we believe your input is crucial. If you feel that these considerations are unnecessary, we are willing to make adjustments to the calculation of sample size at any time.

[P19] Data access was problematic.

The reviewer encountered difficulties when attempting to access the data from the provided link (https://www.wenjuan.pub/s/UZBZJv98ms/). This issue appears to stem from the website's exclusive support for the Chinese language. Using Google Translate, the message displayed on the page was translated to "Collection of this review has been suspended" (该测评已暂停收集) as of October 22, 2023. Despite attempting various options on the webpage, the reviewer was unable to download any files and was consistently redirected to advertisement pages. Hence, the reviewer was unable to obtain any details regarding the authors' translation of the MET. That is, in contrast to their data availability statement, the data is not accessible and prepared for evaluation.

Re: The authors have now uploaded all relevant data to the Open Science Framework and updated the contact information within the manuscript. The video file and questionnaire document for the Chinese version of MET can both be found in the provided link (https://osf.io/kxey3/?view_only=e97871f8520347279c807022e29e381d).

[P21] Data availability is unclear.

The metadata table indicates that the authors have declared, "Yes - any pilot data reported in this submission are fully available, and data collected during the study will be made fully available without restriction upon study completion." However, this protocol mentions a "pre-testing" pilot study involving 20 participants, and yet the data from this pilot study is not available in the submission. This raises concerns about whether this submission is in compliance with the PLOS Data Policy.

Furthermore, it is crucial to include specific details about how the primary dataset will be made available, which should encompass aspects such as obtaining informed consent for data publication and utilizing internationally accessible platforms like the Open Science Framework.

re: Similar to the previous response, the authors affirm the availability of all data. Please access all data files on the Open Science Framework (https://osf.io/kxey3/?view_only=e97871f8520347279c807022e29e381d). The presentation of data has also been adjusted accordingly.

# Minor points #

[P17] The description of the power analysis is unclear, particularly in regard to the test or distribution used to compare Cronbach's alphas. It would be helpful to provide more details on the statistical methods and the specific tests or distributions used in the power analysis. Clarity in describing the statistical procedures and how the power analysis was conducted will enhance the transparency of your research and help readers understand the basis for your sample size determination.

re: Following the recommendation by Ryan (2013), the sample size for this paper was calculated using the Normal Approximation method of the PASS software (PASS 15 Power Analysis and Sample Size Software, 2017, NCSS, Kaysville, Utah, USA, ncss.com/software/pass). We have included additional data in the article to provide a more comprehensive overview. Please refer to the "Data Analysis - Sample Profile" section for further details.

[P19] The description of the data analysis is rather vague. It's not clear which specific tests are being performed to test what relationships or differences, what the expected outcomes are, or what planned contrasts will be used. It would be beneficial to provide more details and clarity regarding the specific statistical tests and analyses that will be conducted, along with the expected outcomes and any planned contrasts or comparisons that are relevant to the study. This will help readers and reviewers better understand the analysis plan and the research objectives.

Re: We sincerely apologize for any inconvenience caused by this issue. To address it, we have completely rewritten and enriched the steps of data analysis, undergoing thorough and multiple proofreading processes throughout the entire manuscript. We hope that these measures effectively resolve the identified issues in the manuscript. 

- It's important to address typos and grammatical errors in your manuscript to ensure clarity and professionalism. Some specific examples are "[P19] Date Analyses" and "[P21] 1830 years." It's advisable to thoroughly proofread and copy-edit the manuscript before submission to ensure that such issues are corrected. This will help enhance the overall quality of the paper.

Re: In response to this issue, we have refined the language and expression throughout the entire manuscript. The text has also been proofread by native speakers for further professionalism.

---

## [Editor Report · Decision Letter 1]

27 Dec 2023

Validation and applicability of the music ear test on a large Chinese sample

PONE-D-23-24643R1

Dear Dr. Xu,

We’re pleased to inform you that your manuscript has been judged scientifically suitable for publication and will be formally accepted for publication once it meets all outstanding technical requirements.

Kind regards,

Hamed Ahmadinia

Academic Editor

PLOS ONE

---

## [Editor Report · Acceptance letter]

12 Jan 2024

PONE-D-23-24643R1 

PLOS ONE

Dear Dr. Xu, 

I'm pleased to inform you that your manuscript has been deemed suitable for publication in PLOS ONE. Congratulations! Your manuscript is now being handed over to our production team.

Kind regards, 

on behalf of

Mr. Hamed Ahmadinia 

Academic Editor

PLOS ONE